# Recent Advances in Selenophene-Based Materials for Organic Solar Cells

**DOI:** 10.3390/ma15227883

**Published:** 2022-11-08

**Authors:** Xuan Liu, Xin Jiang, Kaifeng Wang, Chunyang Miao, Shiming Zhang

**Affiliations:** 1Key Laboratory of Flexible Electronics (KLOFE) & Institute of Advanced Materials (IAM), Jiangsu National Synergetic Innovation Center for Advanced Materials (SICAM), Nanjing Tech University, 30 South Puzhu Road, Nanjing 211816, China; 2Jiangsu Seenbom Flexible Electronics Institute Co., Ltd., Level 2 Building 5, Zhida Road 6, Nanjing 210043, China

**Keywords:** organic solar cells, selenophene, acceptor, donor

## Abstract

Due to the low cost, light weight, semitransparency, good flexibility, and large manufacturing area of organic solar cells (OSCs), OSCs have the opportunity to become the next generation of solar cells in some specific applications. So far, the efficiency of the OSC device has been improved by more than 20%. The optical band gap between the lowest unoccupied molecular orbital (LUMO) level and the highest occupied molecular orbital (HOMO) level is an important factor affecting the performance of the device. Selenophene, a derivative of aromatic pentacyclic thiophene, is easy to polarize, its LUMO energy level is very low, and hence the optical band gap can be reduced. In addition, the selenium atoms in selenophene and other oxygen atoms or sulfur atoms can form an intermolecular interaction, so as to improve the stacking order of the active layer blend film and improve the carrier transport efficiency. This paper introduces the organic solar active layer materials containing selenium benzene in recent years, which can be simply divided into donor materials and acceptor materials. Replacing sulfur atoms with selenium atoms in these materials can effectively reduce the corresponding optical band gap of materials, improve the mutual solubility of donor recipient materials, and ultimately improve the device efficiency. Therefore, the sulfur in thiophene can be completely replaced by selenium or oxygen of the same family, which can be used in the active layer materials of organic solar cells. This article mainly describes the application of selenium instead of sulfur in OSCs.

## 1. Introduction

With the development of the world economy, energy resources are becoming more and more scarce [1], and solar cells, as new energy materials, have a good development prospect in this area. Solar cells have always been a research hotspot, but also a technical problem, which also makes them the focus of attention all over the world. Since American scientists Kearns and Calvin first prepared organic photovoltaic devices with a metal/polymer/metal single-layer device structure in 1958 [2], the various properties of organic solar cells (OSCs) [3] have been greatly improved under the hard research of many scientists. Compared with the production of inorganic solar cells, OSCs have the advantages of a low production cost, light weight, strong design-ability of the organic molecular structure, and easy production processes. They can be produced by spin coating [4], printing [5,6], the or roll-to-roll process [7,8,9,10]. The third-generation solar cell is still in the laboratory stage due to the preparation and selection of the active layer materials, and its efficiency has not yet reached the commercialization level. Therefore, at present, researchers have focused on the research of the third-generation solar cells. If the power conversion efficiency (PCE) can be a further breakthrough, it may be used in production practice as it has terrific application prospects. According to different device structures, organic solar cells can be divided into the following three types: homojunction organic solar cells [11,12,13,14], bulk heterojunction organic solar cells [15,16,17,18], and dye-sensitized nanocrystalline organic solar cells [19,20,21,22,23]. Among them, bulk heterojunction is a thin film composed of electron acceptor (A) materials and electron donor (D) materials. Because a bulk heterojunction OSC [24] has the characteristics of a low manufacturing cost and a high PCE, it has a great value for further research. Figure 1 describes the device structure of a bulk heterojunction OSC. OSCs have different requirements for donors and acceptors. For acceptor materials, they should have a relatively good absorbing ability, a high electron mobility, and a good stability; for donor materials, they should have a high hole mobility, stability, and solubility. A strong intramolecular charge transfer between the donor and the acceptor co-units is preferred, which can broaden the solar absorption spectra and achieve large short-circuit current values (*J*sc) for an OSC application [25]. However, usually scientists prefer to select a donor and acceptor with a complementary absorption. For example, narrow bandgap acceptors can effectively absorb a solar spectrum in a long wavelength, which thus requires the donor to absorb solar light in a short wavelength to result in a complementary absorption. 

Active layer materials, both the acceptor and donor materials, usually contain different aromatic rings to improve the conjugate planarity of the materials, which makes the structural distribution of the blend film more conducive to the carriers transport rate and improve the efficiency of the device [26,27]. Thiophene and selenophene are common building units for designing new conjugated materials. Thiophene is an electron rich aromatic heterocycle ring with sp^2^ hybrid atoms [28] and an excellent stability [29]. Selenium, as a homologous element of the sulfur atom, has a larger atomic radius than sulfur. Hence, selenophene, as a thiophene analogue, is easier to polarize and has a lower aromaticity than thiophene. A building block with the selenophene group can reduce the lowest unoccupied molecular orbital (LUMO) energy level of molecules, resulting in a narrower optical band gap of selenophene containing OSCs materials than that containing a thiophene unit, and finally the light absorption efficiency of the material is improved. According to density functional theory (DFT), we calculated the LUMO and the highest occupied molecular orbital (HOMO) energy levels of furan, thiophene, and selenophene. It is obvious from Figure 2 that selenophene has the smallest band gap. This paper summarized recent the organic active layer materials containing the selenophene groups, including small molecule acceptors, polymer acceptors, small molecule donors, and polymer donors, and discussed their respective advantages and the reasons for their superior properties. The geometry of all the molecules and complexes was optimized through density functional theory (DFT). All the DFT computations were performed by the B3LYP density functional method. The 6–31G(d) basis set was used for the energy calculation of the molecules. All these calculations were performed with the Gaussian 16 software package. Although the existence of selenium can have a force with oxygen or sulfur, it is a good research and development idea. However, the atomic radius of the selenium atom is the largest among the three, which may lead to a steric hindrance, which may hinder the reaction activity in the preparation process, hinder the preparation efficiency, and increase the preparation cost. Therefore, from the perspective of commercialization, thiophene with the second smallest band gap may be the best idea, and selenophene has great prospects in the laboratory stage.

## 2. Selenophene Acceptors for Organic Solar Cells

Fullerenes have been relatively recently developed and have improved acceptor materials, but they also have some disadvantages, such as a limited spectral absorption, a high production cost, and an unstable morphology, which limits their practical application. In recent years, people have invested a lot of effort in the development of non-fullerene acceptors (NFAs). Compared with fullerenes, NFAs have the characteristics of an adjustable energy level, appropriate optical absorption, excellent chemical stability, and a good morphological stability. The small molecule and polymer materials described below belong to non-fullerene acceptor materials.

### 2.1. Selenophene Small Molecule Acceptors

Compared with polymer materials, small molecule materials have the advantages of a simple synthesis, a high purity, and a precise chemical structure. In the structure of small molecules, selenophene has a force with the sulfur atom or oxygen atom next to the selenium atom in the molecule, which can reduce its band gap, improve the surface smoothness of the active layer, and increase the mutual solubility between the small molecular materials and acceptors or donors. According to the advantages of an easy structure adjustment of the small molecular materials, three different positions in the small molecule receptor material: the sulfur atoms on the molecular core, on the bridging groups, and on the ending groups can be replaced by selenium atoms to design corresponding new molecules (Figure 3). 

#### 2.1.1. NFAs with Selenophene as Molecular Core or Side Chain

For NFAs, usually there are three parts: A conjugated core, a side chain, and an ending group. In most cases, NFAs are mainly employing thiophene-fused rings as the center core, while there are relatively fewer cases based on selenophene rings.

Due to a non-covalent interaction being able to enhance the intramolecular charge transfer (ICT) effect, the bandgap of materials with heteroatoms is reduced [30]. Thus, the unit containing heteroatoms, caused by a non-covalent interaction, can extend the absorption spectrum. As a homologous element of sulfur, selenium can replace the sulfur atom on thiophene to produce a new heterocyclic molecule called selenophene. Selenophene has a stronger electron donor ability and a lower aromaticity than thiophene. Therefore, NFAs based on selenophene have a better planarity, a longer conjugated length, and a lower optical band gap. In Table 1, the device parameters of OSCs with selenophene are summarized.

In 2016, Li [31] et al. replaced the sulfur atom in the indacenodithiophene (IDT) structure with the selenium atom, called indacenodithieno [3,2-b] selenophene (IDSe), which was the first time the selenium atom was introduced into the design of the non-fullerene small molecule acceptor’s structure. Because of the π-π* transitions and the existence of ICT, the absorption of the acceptor material IDSe in the thin film was red shifted compared to the solution, and the highest absorption peak in the solid film was about 740 nm. The complementary absorption of poly[(5,6-Difluoro-2-octyl-2H-benzotriazole-4,7-diyl)-2,5-thiophenediyl [4,8-bis [5-(2-hexyldecyl)-2-thienyl]benzo [1,2-b:4,5-b′] dithiophene-2,6-diyl]-2,5-thiophenediyl] (J51) and IDSe-T-IC can enhance the light harvest, so as to increase the *J*_SC_, and a PCE of 8.26% (Table 1) was obtained without any additives. Liu [32] et al. only replaced the sulfur atom on one side of the IDT core with the selenium atom to design an asymmetric molecule 2-(3-oxo-2,3-dihydro-1H-inden-1-ylidene)malononitrile (T-Se). Based on a new core T-Se, fluorine-containing and thiophene-containing end-capping groups were introduced, respectively, to synthesize T-Se-4F and T-Se-Th. The maximum absorption peak of T-Se-Th was red shifted compared with the other two peaks. Atomic force microscopy (AFM) was used to characterize the film morphology of three molecules mixed with the donor poly [[4,8-di [5-(2-ethylhexyl)-4-fluoro-2-thiopheno] benzo [1,2-b:4,5-b′] dithiophene-2,6-diyl]-2,5-thiophene diyl [5,7-di (2-ethylhexyl)-4,8-dioxo-4h, 8h benzo [1,2-c:4,5-c′] dithiophene-1,3-diyl]-2,5-thiophene diyl] (PM6). The T-Se-Th film had the smallest root mean square (RMS) roughness. It was found that the morphology of the T-Se-Th blend was more conducive to a charge transfer. The characterization was consistent with the device results, and the device’s PCE based on PM6:T-Se-Th was 10.29%, while *J*_SC_ and *V*_OC_ reached 16.9 mA/cm^2^ and 0.912 V. Thiophene on one side of the IDT core can also be replaced with selene, and on this basis, an asymmetric center expansion was carried out to obtain the two new molecules SePTT-2F and SePTTT-2F [33]. SePTTT-2F had one more thiophene ring than SePTT-2F, thus SePTTT-2F had a larger conjugation length than SePTT-2F and the long conjugate flatness of SePTTT-2F can improve the LUMO level. The solution absorption and film absorption spectra of the two molecules were close. Finally, due to the improvement of the electron mobility, the PCE of OSCs based on SePTTT-2F can reach 12.24%, higher than that of SePTT-2F (10.09%). Li [34] et al. fused the two IDT cores together with two selenophene rings to produce a novel fused 10-heterocyclic ring. The electron mobility of the fused 10-heterocyclic ring (indacenodithiopheno-indacenodiselenophene) (IDTIDSe-IC) was improved due to the larger selenium atoms, easier polarization, and overlapping molecular orbitals. The ultraviolet (UV) showed that the mixed film of J51:IDTIDSe-IC had a complementary absorption in range of 350–850 nm. The LUMO offset between J51 and IDTIDSe-IC was 0.51 eV and the HOMO offset was 0.12 eV. The mixed films characterized by AFM and transmission electron microscopy (TEM) showed a uniformly dispersed flocculation structure and small nanostructures. An excellent miscibility increased the contact surface of the material, which benefitted the electron mobility (1.27 × 10^−5^ cm^2^/Vs).

Indacenodithienothiophene (IDTT) is a central core extending building block on the basis of IDT [62]. In 2018, a new nuclear SeT was obtained by replacing the sulfur atom on the thiophene group near the phenyl group in the IDT nucleus with the selenium atom [35]. This was an electron-rich fused seven-heterocyclic core, it had a highly planar structure, and it had a stronger electron donating ability than IDTT. Different electron-deficient terminal units were connected by set to obtain the corresponding NFA-SeTIC and SeTIC4Cl. Due to the high electron affinity of the chlorine atom, SeTIC4Cl showed obvious down-shift energy levels and a small optical bandgap. The PCE value of the device was optimized based on SeTIC4Cl:PM6 (13.32%) being higher than that based on the SeTIC:PM6 blend film (7.46%). 

In 2019, Wan [36] et al. connected the benzo [1,2-b:4,5-b] diselenophene-functionalized electron-rich donor core and two chlorinated end groups to synthesize BDSeThCl and BDSePhCl. Compared with the BDSeThCl film, the absorption spectrum of the BDSePhCl film showed a significant red shift of about 100 nm. In addition, according to AFM and TEM, compared to the BDSeThCl:poly[(2,6-(4,8-bis(5-(2-ethylhexyl-3-chloro)thiophen-2-yl)-benzo [1,2-b:4,5-b′]dithiophene))-alt-(5,5-(1′,3′-di-2-thienyl-5′,7′-bis(2-ethylhexyl)benzo [1′,2′-c:4′,5′-c′]dithiophene-4,8-dione)] (PM7) mixture, the BDSePhCl:PM7-based blend film exhibited a better phase separation morphology, a more balanced charge transport, and a weaker charge recombination compared to that of BDSeThCl. These properties make the BDSePhCl:PM7-based blend have relatively higher *J*_SC_ (20.35 mA/cm^2^), fill factor (FF) (0.73), and PCE (13.68%) values. The author also discussed the charge transfer state energy and recombination energy of the materials. According to the calculations, the energy loss caused by a charge recombination in the devices based on BDSePhCl:PM7 was lower than that in devices based on BDSeThCl: PM7. The lesser the energy loss, the lesser the voltage loss, and the higher the EQE of the devices based on BDSePhCl: PM7. In addition, by replacing different dihalogenated end groups, the original benzo [1,2-b: 4,5-b0] diselenophene electron-rich center core (BDSeT) was kept unchanged, and the corresponding small molecule acceptors BDSe-4Cl, BDSe-2 (BrCl), and BDSe-4Br were obtained. The solubility of the three molecules decreased with the increase in the terminal bromine atoms, while the miscibility of the blend film based on the three molecules was gradually improved with the increase in the terminal chlorine atoms. Although the three molecules had different terminal groups, their absorption curves basically coincided. In the device characterization of the three molecules, the FF parameters and the external quantum efficiency (EQE) parameters of BDSe-2(BrCl) were higher than the other two, and its PCE was also the highest, up to 14.54%. In 2021 [38], C8T-BDSe4Cl was synthesized with BDSe as the core and 5,6-dichloro-3-(dicyano methylene) indene-1-one (IC-2Cl) as the end group. Due to the introduction of selenium atoms in the fused core, the *J*_SC_ (21.79 mA/cm^2^) and PCE (13.50%) of the device based on C8T-BDSe4Cl:PM6 were higher than those based on C8T-BDT4Cl:PM6. The two sulfur atoms on the two thiophene groups on the IDTT core were replaced with selenium, and two heterogeneous ladder electron donor blocks were designed. Then, the 5,6-difluoro-3-(dicyanomethylene) inden-1-one (IC-2F) end groups were connected to obtain the two NFA SRID-4F and TRID-4F [39]. The different positions of the selenium atoms in SRID-4F and TRID-4F affected their optical and electrochemical properties, as well as their charge transfer behavior. The selenium atom in SRID-4F was farther away from the central core and had a larger dipole moment, leading to the absorption red shift and the device based on SRID-4F generated a charge transfer. The device based on PBDB-T-2F:SRID-4F showed a PCE of 13.05%, with a *V*_OC_ of 0.846 V, a *J*_SC_ of 20.21 mA/cm^2^, and a 75.2% FF. Hence, the sulfur atom on thiophene near the central phenyl unit was replaced by selenium to construct molecules such as C8T-BDSe-4Cl. On the other hand, the sulfur atom of thiophene far from the central phenyl was substituted by selenium to design molecules such as SRID-4F. A new NFA (TSeTIC) was synthesized [40]. Compared with the thiophene-like molecule (TTTIC), the introduction of the selenium atom led to the up-shift of the HOMO energy level and improved the molecular mobility. In addition, the introduction of the selenium atom also led to the red shift of the corresponding molecular absorption by about 15 nm. The PCE of the optimized PM6: TSeTIC device was 13.71%, which was about 1.66% higher than that of the PM6:TTTIC device. In 2021, Ge [41] et al. replaced the end group with the meta- or para-(2,3-dihydro-3-oxo-1H-inden-1-ylidene)propanedinitrile (INCN) end group relative to dicyanoethylene on the basis of TSeTIC. The different positions of the chloride ions on the end groups lead to a different wavelength of the highest absorption peaks and different absorption intensities of their corresponding molecules. Because IC-M-Cl had a stronger electro-withdrawing power than IC-P-Cl, the absorption of TSeIC-M-Cl was relatively red shifted. The optimized PCE based on the PBT1-C:TSeIC-P-Cl device was 11.26%, which was better than that of PBT1-C:TSeIC-M-Cl (9.72%). Based on IDTT, Liu [42] et al. synthesized SeCT-IC, CSeT-IC, and CTSe-IC. Additionally, they systematically studied the effects of three different positions on the properties of these molecules by changing the positions of the central selenophene in the innermost, and the relatively outer and outermost positions of the central nucleus. The surface of the CTSe-IC and J71 mixed membrane was rough and had an obvious phase separation. These characterization results were consistent with the higher charge mobility of CTSe-IC. Therefore, the charge mobility greatly made up for the shortcomings of CTSe-IC. Finally, the PCE of the devices based on CTSe-IC reached 11.59%, which was much higher than the other two. The PCE of the devices based on SeCT-IC and CSeT-IC were 10.89% and 8.52%. The *V*_OC_ of devices based on SeCT-IC, CSeT-IC, and CTSe-IC were 0.889 V, 0.912 V, and 0.925 V. The *J*_SC_ of the devices based on the three molecules were 18.09 mA/cm^2^, 17.17 mA/cm^2^, and 18.21 mA/cm^2^. The FF of the devices based on the three molecules were 67.69%, 54.83%, and 68.81%, respectively.

Another classical NFA is 12,13-bis (2-ethylhexyl)-3,9-bis (DECA)-12,13-dihydro-[1,2,5] thiadiazo [3,4-e] thiopheno [2″,3″: 4′,5′] thiopheno [2′,3′: 4,5] pyrrolo [3,2-g] thiopheno [2′,3′: 4,5] thiopheno [3,2-b] indole-2,10-bis (5,6-difluoro-3- (dicyano methylene) indene-1-one) (Y6). Y6 is composed of a ladder-type electron-deficient-core-based central fused ring dithienothiophen [3,2-b]-pyrrolobenzothiadiazole and a benzothiazole (BT) core. This structure has the benefits of a good absorption and electron affinity [63]. Based on Y6, Chai [43] et al. designed and synthesized a series of NFA BPF-4F- and BPS-4F-containing furan and selenophene groups. Due to the high polarizability of selenium, BPS-4F had a better absorption and a stronger red-shifted absorption than BPF-4F and Y6. The *J*_SC_ based on the BPS-4F device was also the highest among the three, reaching 25.4 mA/cm^2^. However, the EQE spectral curve of the BPS-4F device was slightly lower than the other two. The PCE value of BPS-4F was 0.5% lower than that of the BPT-4F-based devices, but 4.2% higher than that of the BPF-4F-based devices. On the basis of Y6, it not only replaced the sulfur atom on thiophene, but also increased the carbon atom in the inner chain from 8 to 11 to obtain a new molecule CH1007 [44]. These changes promoted the red-shifted optical absorption of CH1007. As the sulfur atom in thiophene was replaced by selenium, the intramolecular and intermolecular interactions were strengthened and the molecular properties were adjusted. The experimental results showed that the *J*_SC_ of the PM6: CH1007:PC_71_BM device was 27.48 mA/cm^2^, and the PCE was as high as 17.08%. Unlike CH1007, S-WSeSe-Cl designed by Yang [45] et al. used chlorine molecules to replace the four fluorine molecules at the end group. A-SWSe-Cl was different from S-WSeSe-Cl in that A-SWSe-Cl was unilateral sulfur replaced by selenium. The experimental results showed that with the increasing amount of selenium in the nucleus, the optical band gap became narrower, the intermolecular force became stronger, the maximum absorption peak increased, and the absorption curve red shifted. According to the characterization of AFM and TEM, the surface of the PM6: A-SWSe-Cl blend film was rougher, which increased the area of the charge transport interface and contributed to the exciton dissociation and charge transport. Therefore, the PCE of the asymmetric A-SWSe-Cl was about 1.5% higher than that of S-WSeSe-Cl. This was the highest PCE of the devices containing selenium materials at present. We have reason to believe that if the side chains of the materials are increased to improve the solubility, or donor materials that are more suitable for LUMO and HOMO are selected, the PCE will increase more. The three molecules designed by Kim [46] et al., YSe-C3, YSe-C6, and YSe-C9, were compared by shortening the length of the side chain. N-propyl (C3), N-hexyl (C6), and N-nonyl (C9) were used as the side chain of the molecule, respectively. Among the three mixture films, PM6: YSe-C6 had the lowest root mean square (RMS) roughness, so the morphology of its blend film was the best. The EQE value of the PM6: YSe-C6 device was the best among the three, and the FF (0.73) based on the PM6: YSe-C6 device was higher than the other two devices. Therefore, the PCE of the PM6: YSe-C6 device was the highest among the three, which reached 16.11%, and this was higher than the PM6: YSe-C3 device (11.63%) and the PM6: YSe-C9 device (14.20%). Qi [47] et al. designed and synthesized a series of benzotriazole (Bz) fused ring NFAS. By introducing different linear N-alkyl and branched N-alkyl, four molecules, mBzS-4F, PN6SBO-4F, AN6SBO-4F, and EHN6SEH-4F, were obtained. Since the N-methyl alkyl group had the shortest length among the four molecules, mBzS-4F had a strong face-to-face π-core interaction; although 2-ethylhexyl had a long length, it had a strong intermolecular interaction in the adjacent molecular end groups and still formed an effective charge-transport channel. Therefore, the OSCs based on PM6: mBzS-4F and PM6:EHN6SEH-4F showed very high PCEs of 17.02% and 17.48%, respectively. In 2021, Qi [48] et al. also designed a new NFA molecule, EHBzS-4F, by increasing the number of carbon atoms in the inner side chain. Due to the presence of selenophene in the core, the intra- and intermolecular interactions increased, which made the absorption edge of EHBzS-4F be located at 996 nm. However, because the branched N-alkyl chain of the Bz part had a steric effect, which undermined the morphology of the blend, the PCE based on PM6: EHBzS-4F was 15.94%. There was a sulfur atom at the top of the benzothiadiazole in the Y6 structure, which can be replaced to obtain a new molecule Y6Se [49]. Due to the oxidation resistance of the organic selenides, Y6Se had a better intrinsic photo-stability than Y6 under strong UV light. According to the UV absorption spectrum, it was found that the absorption of the selenium-containing molecule Y6Se was about 25 nm red shifted than that of the selenium-free molecule Y6. The surface roughness of the D18:Y6Se and D18:Y6 blend films was 1.8 and 0.7 nm, respectively, and D18:Y6Se had a distinct fiber network interpenetrating structure of the optical fiber network. The experimental results showed that the PCE based on the D18:Y6Se device was as high as 17.7%. Yu [50] et al. replaced the sulfur in the thiophene part of the Y6 molecule with selenium to synthesize Y6-2Se, and compared the performance with Y6Se to study the effect of different substitution positions of selenium in the molecule on the molecular properties. The experiments showed that the selenium substitution on the benzothiadiazole ring was more effective than the selenium substitution on thiophene, which improved the compact molecular packing and absorption red shift of the solid molecules, therefore, the PCE based on the PM6:Y6Se (16.02%) device was higher than that based on PM6:Y6-2Se (14.94%). Song [51] et al. introduced a vinylene π-bridge into symmetrical molecules. By increasing the conjugation length and introducing an alkyl-substituted heterocyclic π-bridge on the conjugated main chain, the bandgap was reduced and the ICT was enhanced. Selenium atoms with a stronger polarization than sulfur were introduced, which was conducive to improving the charge transport. The poly ([2,6′-4,8-bis-((2-ethylhexyl)-thiophene-5-yl) benzo [1,2-b; 3,3-b] dithiophene] -alt-[1,3-bis-(thiophene-5-yl)-5,7-bis-(2-ethylhexyl) benzo [1,2-c:4,5-c′] dithiophene-4,8-dione]) (PBDB-T):BTP-Se blend film had a good absorption and a low energy loss, which indicated that the synergistic effect of the vinylene π-bridge and selenium fused-heterocycle core can effectively reduce the energy loss (E_loss_) of the devices. The EQE value of the PBDB-T:BTP-Se blend film (83%) was also the highest, which indicated that this structure can effectively improve the EQE.

In addition to the above classical molecular frameworks, there were some NFAs of other structures that had been made by people. Six thiophenes were fused together to synthesize thiophene-thieno [3,2-b] thiophene-thiophene (4T). The thiophene sulfur on both sides of the 4T group was replaced by selenium to obtain selenophenol thiophene [3,2-b] thiophene selenophenol (ST) [52]. Then, it was connected with the 3- (dicyano methylene) indene-1-one (IC) terminal group to obtain the STIC molecule. Compared with the unsubstituted molecular 4TIC, the STIC film’s absorption was red shifted about 50 nm. The EQE curve of the devices based on PBDB-T:4TIC was significantly higher than that based on PBDB-T: STIC, which was consistent with the *J*_SC_ parameters of the devices. Three other NFAs were designed and synthesized by Tang [53] et al., two symmetrical molecules MQ3 and MQ5, and an asymmetric molecule MQ6 based on different heteroheptacenes, while the same end group IC-2F was used. The three molecules all contained selenophene groups. The difference was in the position and number of selenophene rings. The molecular MQ6, containing only one selenophene, was asymmetric, showing a strong π-π stacking and good carrier transport properties, as well as a large dipole moment. Therefore, the devices based on PM6:MQ6 had the highest *J*_SC_ (24.62 mA/cm^2^) and the highest PCE of 16.39%.

Gao [54] et al. used n-hexyl selenophenyl as a side chain connecting to the IDTT core, and 2-(6-oxo-5,6-dihydro-4h-cyclopenta [c] thiophene-4-ylidene) malononitrile (CPTCN) as end group to prepare ITCPTC-Se. N-hexyl selenophenyl had an inductive effect, which increased the electric density of the molecule backbone, so the absorption performance of ITCPTC-Se was improved. The HOMO and LUMO orbitals of ITCPTC-Se were evenly distributed along the whole backbone, which was conducive to the hole and electron transport in the molecule. The maximum ultraviolet absorption peak of ITCPTC-Se was significantly higher than that of ITCPTC-Th, but the device parameters based on poly ([2,6′-4,8-bis-((2-ethylhexyl)-thiophene-5-yl) benzo [1,2-b; 3,3-b] dithiophene] -alt-[1,3-bis-(thiophene-5-yl)-5,7-bis-(2-ethylhexyl) benzo [1,2-c:4,5-c′] dithiophene-4,8-dione]) (PBDB-T):ITCPTC-Se, such as the *J*_SC_ (15.20 mA/cm^2^), FF (0.683), and PCE (9.02%), were slightly lower than those based on PBDB-T:ITCPTC-Th.

#### 2.1.2. NFAs with Selenophene as Bridging Groups

To improve the properties of the small molecule acceptors, scientists also introduced bridging groups. Extending the thiophene or other groups around the bridged carbon atom can increase the conjugation and crystallinity, so as to form a more densely compact and orderly stacking [64]. By replacing the sulfur in the classical bridging group thiophene with selenium, many new molecules were designed. Liang [55] et al. replaced the thiophene bridge in 2,2′-((2Z,2′Z)-((4,4,9,9-tetrahexyl-4,9-dihydro-s-indaceno [1,2-b:5,6-b′]dithiophene-2,7-diyl)bis(methanylylidene))bis(3-oxo-2,3-dihydro-1H-indene-2,1-diylidene))dimalononitrile (IDIC) and IDIC-4F with selenophene to synthesize two new NFAs, named IDT2Se and IDT2Se-4F. According to the ultraviolet-vis-near infrared absorption spectra of the two molecules in the solution and films, IDT2Se-4F showed an absorption redshift compared with IDT2Se, which meant that the interaction between the IDT2Se-4F molecules was stronger. The *J*_SC_ (25.49 mA/cm^2^) and FF (0.66) of the devices based on PBDB-T: IDT2Se-4F were higher than those based on PBDB-T:IDT2Se, so the PCE of the former was higher, up to 11.19%. With the IDT as the core, selenium benzene as the bridge group, and IC, IC-F, and IC-2F as the terminal groups, three kinds of NFAS were designed: IDT2SeC2C4, IDT2SeC2 C4e2F, and IDT2SeC4e4F [56]. The side chain of the IDT adopts alkyl chain molecules. Unlike the phenyl side chain, the alkyl chain has a more orderly and closer packing, so their electron mobility is higher. Depending on the fluorine content in the end group, IDT2SeC2C4-2F and IDT2SeC2C4-4F showed a red-shifted absorption, compared to IDT2SeC2C4. The devices based on IDT2SeC2C4-2F and IDT2SeC2C4-4F also had a higher PCE (10.20% and 10.57%) than IDT2SeC2C4 (8.92%). Liu [30] et al. designed three kinds of NFAs: IDTO-T-4F, IDTOSe-4F, and IDTO-TT-4F. These three NFAs were synthesized with 4,4,9,9-tetrakis(4-hexylphenyl)-3,8-bis(octyloxy)-3a,4,9,10b-tetrahydro-s-indaceno [1,2-b:5,6-b′] dithiophene (IDTO) as the core, thiophene, selenophene, and thieno [3,2-b] thiophene as the bridging group, and IC-2F as the end group. Similar to the 4TO-T-4F and 4TO-Se-4F molecules, there was also a non-covalent interaction between oxygen, sulfur, or selenium in these three molecules. In the range of 600–900 nm, the three NFAS had a strong light absorption. The film of IDTO-Se-4F had a red shift relative to the other two molecules’ absorption. When PBDB-T was used as the donor material, the devices based on three NFAs IDTO-T-4F, IDTOSe-4F, and IDTO-TT-4F obtained a high PCE of 12.62%, 10.67%, and 10.21%, respectively. The new molecules MPU1 and MPU4 could be synthesized by linking diketopyrrolopyrrole (DPP), thiophene, or selenium with rhodanine [57]. These two molecules had a high planarity and conjugation length. Due to the existence of selenium atoms with a high polarizability in MPU4, MPU4s absorption exhibited a great red shift to MPU1, which indicated that the interaction between the MPU4 molecules was stronger and the electron delocalization on the MPU4 skeleton was stronger than that of MPU1. The optimal PCE (10.05%) of the devices based on the ternary active layer SM1: PC_71_BM: MPU4 was higher than that based on SM1: MPU4 (8.96%) and SM1:MPU1 (7.22%), and its EQE curve was also much higher than that of the other two devices.

Compared with fused ring molecules, non-fused ring NFA molecules have the advantages of a simple synthesis. Ding [58] et al. designed and synthesized a simple non-fused ring acceptor 2T2Se-F, which was obtained by connecting 2,6-di(hexyloxy) phenyl-substituted biothiophene and the terminal groups of fluorinated 1,1-dicyanomethylene-3-indone (DFIC). The existence of selenophene made the absorption red shifted and the energy level rise, resulting in strong intramolecular and intermolecular interactions. Based on the PM6:2T2Se-F device, after adding 1,8-diio-dooctane (DIO) and thermal annealing, a high PCE of 12.17% was obtained because of its high crystallinity, high mobility, and good morphology.

### 2.2. Polymer Acceptors with Selenophene

In 1993, Sariciftci and his co-workers [65] prepared a double-layer heterojunction solar cell based on the C_60_ derivative PCBM and MEH-PPV, with a conversion efficiency of 2.9%. Since then, polymer solar cells (PSCs) have become a research hotspot. Compared with traditional inorganic semiconductor solar cells, PSCs have the outstanding advantages of being light, thin, flexible, solution processable, etc.

In 2021, Fan [59] et al. designed three novel polymerized acceptors with a fused selenophene-containing main chain, namely PFY-1Se, PFY-2Se, and PFY-3Se (Figure 4). With the increase in the selenium content in the molecule, the absorption of the polymer acceptor and polymer donor PBDB-T was more and more complementary. Compared with the selenium-free molecule PFY-0Se, PFY-3Se, containing selenophene, had a lower Eg (1.449 eV). All the polymer solar cells based on PBDB-T: PFY-3Se achieved a high PCE over 15% with a low energy loss under the condition of a high *J*_SC_ (23.6 mA/cm^2^) and FF (0.737). Fan [60] et al. designed PY2Se-F and PY2Se-Cl molecules by connecting a halogen atom to the repeating unit thiophene based on the PFY-2Se molecule. Fluorination can improve the electron absorption intensity of the end group, and the spatial effect of the chlorine atom can reduce the co-planarity of the conjugated main chain. Therefore, compared with selenium free PY2S-H and fluorine-containing PY2S-F, PY2Se-F and PY2Se-Cl showed a significant red shifted absorption. When mixed with the polymer donor PM6, the PCE of PY2S-H, PY2S-F, PY2Se-F, and PY2Se-Cl gradually increased from 14.8% to 16.1%. The PCE of the device based on PY2S-H was 14.8%, the *J*_SC_ was 22.3 mA/cm^2^, the *V*_OC_ was 0.94 V, and the FF was 0.71. The PCE of the device based on PY2S-F was 15.1%, the *J*_SC_ was 23.3 mA/cm^2^, the *V*_OC_ was 0.92 V, and the FF was 0.71. The PCE of the device based on PY2Se-F was 15.6%, the *J*_SC_ was 14.4 mA/cm^2^, the *V*_OC_ was 0.89 V, and the FF was 0.72. The PCE of the device based on PY2Se-Cl was 16.1%, the *J*_SC_ was 24.5 mA/cm^2^, the *V*_OC_ was 0.89 V, and the FF was 0.74.

Different from the above building blocks, Lee [61] et al. designed a series of new n-type semiconductor polymers with an A-D_1_-A-D_2_ structure. P(NDI2OD-Se-Th_x_) had controllable crystal properties, which can be applied to all PSCs. By changing the content of selenium, thorium, and Se-Th in the polymer acceptors skeleton, the absorption range and crystallinity of the material can be reasonably controlled. With the increase in the Se-Th content in the polymer molecules, the absorption of the corresponding polymer molecules was more red shifted, and the absorption range was wider. The addition of Se-Th in the main chain of P(NDI2OD-Se-Th_0.8_) successfully controlled the crystallinity of the polymer acceptors and formed a good blend morphology with PBDB-T. The PCE of P(NDI2OD-Se-Th0.8) was the highest, reaching 8.30%, the *J*_SC_ was 14.2 mA/cm^2^, the *V*_OC_ was 0.88 V, and the FF was 0.66.

## 3. Selenophene Donors in Organic Solar Cells

The OSC device generates excitons after a light absorption. After the excitons are dissociated, the electrons are transferred from the donor material to the acceptor’s material, the holes and electrons are transferred in the donor’s material and the acceptor’s material, respectively, and they are finally absorbed by the electrode. Therefore, the selection of donor materials in organic solar energy has a great impact on its device efficiency. Like the acceptor materials, donor materials can be simply divided into polymer materials and small molecule materials.

### 3.1. Small Molecule Donors with Selenophene

Small molecular materials have the advantages of a high purity and a small batch difference and they are widely used in solar cell donor materials. The optical and electrochemical properties of small molecule donors can be adjusted by introducing different donors, acceptors, and bridging groups.

Xu [66] et al. designed and synthesized small molecular donors of thiophene or selenophene substituents, L1 (Figure 5) of the thiophene ring and L2 of the selenophene unit on the benzo [1,2-b: 4,5-b0] dithiophene (BDT) central group (Figure 5). The main difference between L1 and L2 was the conjugated side chain of BDT. Due to the existence of selenium in L2, it had a stronger intramolecular interaction, which led to a better morphological distribution and a lower charge recombination. The UV absorption of the L1 film was slightly wider than that of the L2 film. The PCE of the device based on L2:Y6 was as high as 15.8% (Table 2). More importantly, when the thickness of the active layer of L2:Y6 was up to 300 nm, the PCE was 14.3%, the *V*_OC_ was 0.83 V, the *J*_SC_ was 26.35 mA/cm^2^, and the FF was 0.72. These data were higher than those of the devices based on L1:Y6: the PCE of the devices based on L1:Y6 was 14.6%, the *V*_OC_ was 0.83 V, the *J*_SC_ was 25.28 mA/cm^2^, and the FF was 0.70.

### 3.2. Polymer Donors with Selenophene

#### 3.2.1. Polymer Donors with Selenophene as Bridge

Because there are already many polymers containing thiophene rings, many researchers could design new polymers by replacing thiophene with selenophene easily. The structure of polymer donors with selenophene as the bridge is showed in Figure 5. Two new polymer donors VC6 and VC7 were prepared by connecting the diketopropyrrole core and the Zn porphyrin peripheral moieties with thiophene and selenophene, respectively [67]. Due to the introduction of the selenium atoms, the band gap (1.4 eV) of VC7 was significantly smaller than that of VC6 (1.72 eV) due to the stronger electron donating ability of selenophene. Although, the UV absorption curves of both the solutions and the films were basically consistent. However, the VC7-based devices had a higher PCE (9.24%), *J*_SC_ (15.98 mA/cm^2^), and FF (0.66). The polymer PCDSeBT with dibenzothiadiazole (DSeBT) and 2,7-carbazole building blocks can be prepared by Suzuki coupling polymerization [68]. AFM was used to compare the surface morphology of the PCDSeBT: PC_71_BM mixed film before and after thermal annealing. It was found that the root mean square roughness decreased by 0.04 nm after thermal annealing. According to the comparison of the device performance data, the *J*_SC_ (11.7 mA/cm^2^), FF (0.45), *V*_OC_ (0.79 V), and PCE (4.12%) of the OSCs based on PCDSeBT: PC71BM had been significantly improved after thermal annealing. In a word, this may be because the surface roughness of the films was reduced after thermal annealing, and the charge separation and transmission became more favorable. Gao [69] et al. designed two new D-π-A polymers, PSeBDDIDT and PSeBDDIDT, with an IDT unit and benzo [1,2-c:4,5-c′]dithiophene-4,8-dione (BDD). Comparing the polymer PThBDDIDT with the thiophene unit, the solution and the film of the polymers PSeBDDIDT and PTzBDDIDT showed a red shift absorption, and the former PSeBDDIDT had more red shift. The strong red shift of PSeBDDIDT can improve the photon acquisition rate, resulting in a high current. The corresponding devices were prepared by mixing the three polymer donors with the acceptor PC_71_BM. The *J*_SC_ (16.04 mA/cm^2^) and PCE (8.65%) of the devices based on PSeBDDIDT: PC_71_BM were much higher than the other two, which may be due to the larger absorption range and better molecular flatness of the PSeBDDIDT polymer. It was also possible to use m-alkoxyphenyl-substituted benzodithiophene (BDT-m-OP) as a donor unit and BDD as an acceptor unit. Two polymers with thiophene and selenophenol as the π bridge units were designed [70]. Compared with PBPD-Th, the PBPD-Se containing selenophene not only absorbed the red shift, but also had a higher EQE value and a wider EQE spectrum based on the PBPD-Se: PC_71_BM device. These results were consistent with the *J*_SC_ and PCE results of the device. After adding 1% DIO as an additive, the PCE of the PBPD-Se:PC_71_BM device can be as high as 9.8%, which is 1.4% higher than that of PBPD-Th:PC_71_BM. Zhang [71] et al. prepared PHI-Se-containing selenophene and PHI-Th-containing thiophene based on BDT and phthalimide (PHI). The performance differences of the binary, ternary, and quaternary organic solar devices were studied by using the donor PM6 and the acceptors Y6 and PC_71_BM. 

For two-component devices, the PCE (10.5%) of the devices based on PhI-Se and PhI-Th was relatively close. For the three-components OSCs based on PhI-Th: PM6: Y6 and PhI-Se: PM6: Y6, the maximum PCE was 16.4% and 15.6%, respectively. By adding PhI-Se to the PM6:Y6:PC_71_BM mixture to prepare the quaternary organic solar cells, when the weight ratio of PhI-Se was 0.15, the maximum PCE of 17.2% could be obtained, and the *J*_SC_, *V*_OC_, and FF were 26.3 mA/cm^2^, 0.851 V, and 0.77. Cao [72] et al. designed two kinds of six-ring molecules, TD1 and TD2, to synthesize four D-A polymers: PThTD1, PSeTD1, PThTD2, and PSeTD2. From PThTD1 to PSeTD1, PThTD2, and PSeTD2, the PCE of the devices based on its polymer gradually increased. The PCE of the OSC based on PSeTD2 with a shorter side chain was 1.96% higher than that of PSeTD1 with a longer side chain. This was because selenophene polymers had a better lamellar packing than thiophene polymers, which can be found in the X-ray diffraction (XRD) diagram. The shorter side chain can facilitate the filling of the main chain and improve the hole mobility. P (Se) was synthesized by the copolymerization of selenol, thiophene pyrrolidone (TPD), and thiophene with an alkyl chain [73]. This polymer had a more red-shifted absorption and a wider absorption range than that of P(S) without selenophene. According to the CV test, the HOMO level of P (Se) was relatively deep, which led to a higher *V*_OC_ of 0.88 V. After adding 1-chloronaphthalene (CN) as an additive to the organic solar cell based on P(Se): PC_71_BM, it was found that due to the decrease in the FF and the increase in the series resistance, the oxidation degradation degree of the device with the additive CN was higher than that of the device without the additive. This meant that the designed P(Se) can be well used in additive-free organic solar devices. 

Zhao [74] et al. introduced selenium atoms on the basis of the PTEI-T polymer to study the effect of the non-covalent interaction between O···S and O···Se on the performance of organic solar cells. The introduction of selenium reduced its optical band gap. According to the UV spectrum, the absorption of the films of the two selenium-containing PTEI-S and PSEI-T polymers became wider and PTEI-S was significantly red shifted about 23 nm. According to DFT, the distance between the carbonyl oxygen atom and the sulfur atom or selenium atom in PTEI-S and PSEI-T was much less than the sum of the two van der Waals radius. This indicated that there were non-covalent interactions between the sulfur or selenium and the carbonyl oxygen between the two polymers, which led to the co-planarity of the main chain of the polymer. Dou [75] et al. replaced the sulfur atom on DPP with the selenium atom and copolymerized with thienylbenzodithiophene (BDTT). It was found that PBDTT-SeDPP with a narrow band gap and a strong charge transfer could be obtained. Compared with the thiophene-containing and oxole-containing polymers, the absorption spectrum of PBDTT-SeDPP was significantly red shifted about 50 nm, and the absorption rate was very high in the near-infrared region. Selenium was more easily oxidized than sulfur or oxygen, so the electrons of selenophene were relatively stable, resulting in a low bandgap. The PCE of the PBDTT-SeDPP-based single junction PSCs was 7.2%.

#### 3.2.2. Polymer Donor with Selenophene as Side Chain

In addition to the above polymers containing selenophene structural units, there are also some polymers in which the sulfur atom of thiophene in other positions is replaced by selenium. A new selenium-containing copolymer PDTP-BDTSe can be obtained by replacing thiophene with selenophene on the side chain of the BDT unit [76]. The dithieno [3,2-b:2′,3′-d] phosphole oxide (DTP) unit in the copolymer was highly polarized, which made more charge carriers transfer to the acceptors, resulting in a larger *J*_SC_ (14.8 mA/cm^2^). By introducing 0.5 vol% 1,8-octanedithiol (ODT) as an additive to the devices based on PDTP-BDTSe: PC_71_ BM, the stacking of the PDTP-BDTSe polymer chains became more orderly, therefore, the maximum PCE increased from 5.52% to 7.08%. Fluorine substituted benzothiadiazoles change the number of fluorine atoms on benzothiazole, and then connect selenium benzothiophene through still polycondensation reaction., and obtained three new molecules: PBDTSe-BT, PBDTSe-FBT, and PBDTSe-FFBT [77]. By mixing the device with the polymer receptor PC_71_BM, the data feedback obtained that the PCE of the PBDTSe-FFBT containing two fluorine atoms was about 2.63% at most. Chang [78] et al. copolymerized the selenophene-substituted BDT molecule with 2-ethylhexyl-4,6-dibromo-3-fluorothieno [3,4-b] thiophene-2carboxylate (TT) to synthesize PBDTSe-TT. Through the TEM characterization, it was found that the nanostructure of the PTB7Th:PC71BM film had a great phase separation when DIO was not added; after adding DIO, the morphology was optimized. On the contrary, the film morphology of PBDTSe-TT: PC71BM was optimized before adding the additives. After adding DIO, the charge dissociation and charge transfer efficiency decreased. Therefore, the PCE of the devices based on PBDTSe-TT: PC71BM can reach 8.8% without additives. Not only can the sulfur atom on the BDT unit be replaced, Liu [79] et al. also inserted the sulfur atom on the alkyl branched chain to synthesize PBDTS-Se-TAZ and PBDT-Se-TAZ. From the solution to solid film, the absorption of PBDTS-Se-TAZ and PBDT-Se-TAZ had red shifted, which was because the molecular aggregation in the solid film was enhanced. According to the CV test, the HOMO and LUMO energy levels of PBDTS-Se-TAZ decreased, but this result had a weak effect on the band gap; the final device test results showed that the *V*_OC_ value (0.84 V) of the devices based on PBDTS-Se-TAZ: ITIC was higher. The PCE of the PBDTS-Se-TAZ-based device (12.31%) was higher than that of the PBDT-Se-TAZ-based device (10.07%).

The sulfur on the side chain of the BDT unit can be replaced not only by selenium, but also by oxygen to explore its impact on the properties of the polymers. The size of the three atoms was in the order of O < S < Se, and the angle between them and adjacent carbon atoms decreased as the size increased. The structural distortion of the sulfur-containing and selenium-containing polymers was greater than that of the oxygen-containing polymers. This [80] resulted in a more distorted structure of BDT (T) and BDT (S) relative to BDT (F). This affected its optical properties, and the PCE of the BDT (T) (6.5%) and BDT (S) (4.7%) was significantly higher than that of the BDT (F) (3.0%). Intemann [81] et al. used selenophene to replace thiophene in the core, and then copolymerized with 4,7-dibromo-5,6difluoro-[1–3] benzothiadiazole to obtain a new polymer, PIDSe-DFBT. Compared with the original PIDT-DFBT (1.88 eV), the optical gap of PIDSe-DFBT (1.80 eV) was smaller. In addition, selenium atoms increased the extinction coefficient of the polymer films. According to the device test, the *J*_SC_ (13.7 mA/cm^2^), FF (0.56), and PCE (6.79%) of the devices based on PIDSe-DFBT were larger than those based on PIDT-DFBT, its *J*_SC_ was 11.2 mA/cm^2^, the FF was 0.55, and the PCE was 6.02%.

In summary, there are two methods to replace sulfur atoms, one is to replace the sulfur in thiophene as a separate structural unit and the other is to replace the sulfur in thiophene as a part of the structural unit. Zhong [82] et al. designed two polymers, J75 and J76, on the basis of classical polymer J71. The absorption curves of the selenium-substituted J75 and J71 were very close, while J76 was relatively red shifted. According to the CV test, the selenium substitution of the bridging group units reduced the optical band gap compared with the branched chain substitution. Then, the three polymers were mixed with fullerene acceptors and non-fullerene acceptors to explore their device performance. The results showed that the devices based on J76:PC71BM and J71:m-ITIC had a better performance: their PCEs were 8.37% and 11.73%. This showed that a selenium substitution on the bridging group can improve the PCE of the fullerene acceptor-based PSCs, but this method had a weak impact on the photovoltaic performance of non-fullerene acceptor-based PSCs. 

## 4. Summary and Prospect

The PCE of the organic solar cells containing selenium atoms mentioned in this report is close to 18%. The active layer materials containing selenium have interesting properties, such as a large electron cloud, a strong intramolecular charge transfer ability, a high HOMO energy level, and a good molecular order and crystallinity. These characteristics can improve the short current density, the filling factor, and finally the PCE of organic solar cell devices.

However, the materials containing selenophene may have some poor film morphology or an uneven mixing of the donor materials and acceptor materials. In addition, the electron donating ability and skeleton flatness of the selenophene materials are slightly weaker than those containing thiophene and oxole, with a strong intramolecular charge transfer effect, which is the shortcoming of the selenium-containing active layer materials. According to these deficiencies, adjustments can be made from the following points: (1) the best matching donor/acceptor materials are selected according to the LUMO and HOMO energy levels of the selenium-containing molecules, and the possible interaction of heteroatoms between the molecules when the two materials are stacked should be considered; (2) adjust the position of the selenium atom in the molecule, that is, while introducing the selenium atom, pay attention to the interaction of the original O···S in the molecule, especially some molecules with IC and its derivatives as end groups; (3) the low solubility of the mixed membrane caused by the introduction of the selenium atoms can be compensated by replacing the end group branches. This is because different branched and straight chains can not only significantly improve the solubility of the corresponding molecules, but also improve the mixing degree with another molecule; and (4) by increasing the amount of thiophene in the molecule, the disadvantage that the skeleton flatness is not as good as other molecules due to the introduction of selenophene can be improved, that is, the conjugation ability of the molecules can be improved. With the increase in the conjugation ability, the flatness and stacking order of the molecular skeleton are naturally improved.

Through the continuous efforts of researchers, the properties of the active layer materials with selenophene will be continuously improved, and its application in the field of OSCs will be more and more widely.

## Figures and Tables

**Figure 1 materials-15-07883-f001:**
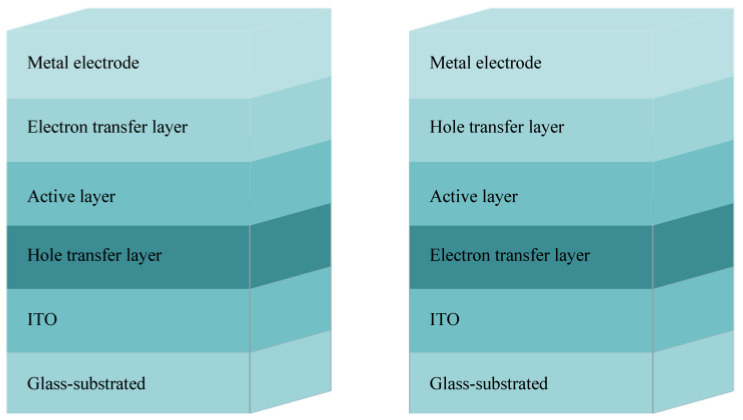
Schematic diagram of organic solar cell structure.

**Figure 2 materials-15-07883-f002:**
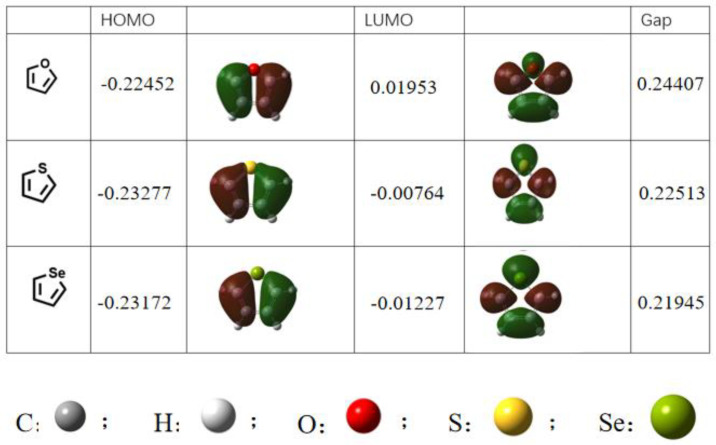
DFT for furan, thiophene, and selenophene.

**Figure 3 materials-15-07883-f003:**
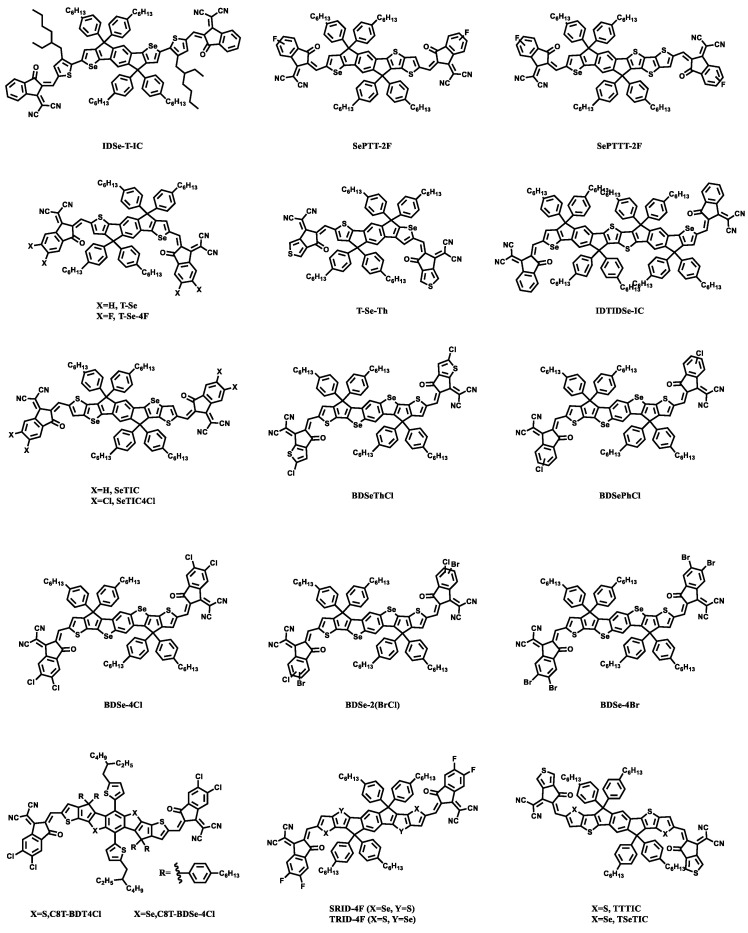
Selenophene small molecule acceptors.

**Figure 4 materials-15-07883-f004:**
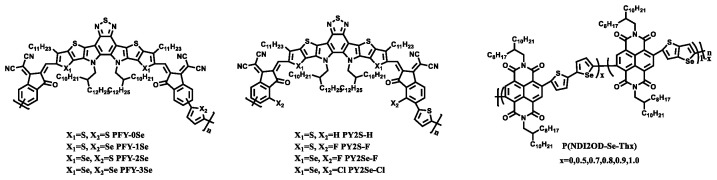
Polymer acceptors with selenophene.

**Figure 5 materials-15-07883-f005:**
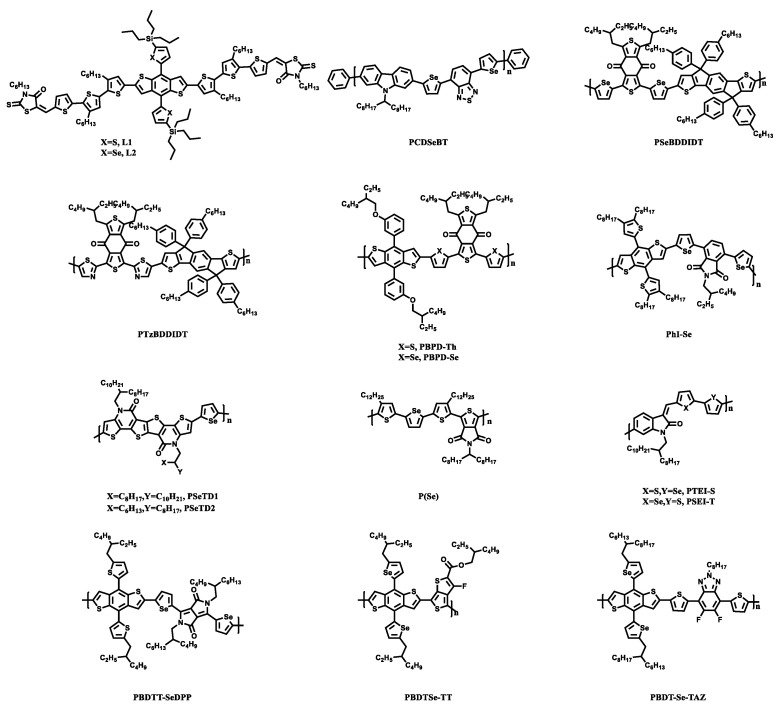
Donors substituted by selenophene.

**Table 1 materials-15-07883-t001:** Device parameters for selenophene-substituted acceptors OSCs.

Acceptor	Ref.	E_g_^opt^(eV)	HOMO/ LUMO (eV)	Voc (V)	J_max_/ mA^−2^	FF (%)	PCE (%)	Donor
IDSe-T-IC	[31]	1.52	−5.45/−3.79	0.91	15.20	62.0	8.58	J51
T-Se	[32]	1.62	−5.61/−3.83	0.94	13.32	61.9	7.36	PM6
T-Se-4F	[32]	1.58	−5.62/−3.93	0.78	18.16	67.3	9.50	PM6
T-Se-Th	[32]	1.60	−5.61/−3.88	0.91	16.99	67.1	10.34	PM6
SePTT-2F	[33]	1.50	−5.71/−4.00	0.83	17.51	75.0	10.90	PBT1-C
SePTTT-2F	[33]	1.50	−5.66/−3.97	0.90	18.02	75.9	12.24	PBT1-C
IDTIDSe-IC	[34]	1.52	−5.41/−3.81	0.91	15.16	58.0	8.02	J51
SeTIC	[35]	1.58	−5.55/−3.90	0.95	15.45	51.0	7.46	PM6
SeTIC4Cl	[35]	1.44	−5.65/−4.08	0.78	22.92	75.0	13.32	PM6
BDSeThCl	[36]	1.55	−5.58/−3.82	0.97	17.85	68.8	11.91	PM7
BDSePhCl	[36]	1.41	−5.61/−3.96	0.92	20.35	73.1	13.68	PM7
BDSe-4Cl	[37]	1.39	−5.63/−3.94	0.83	22.50	74.1	13.83	PM7
BDSe-2(BrCl)	[37]	1.39	−5.64/−3.95	0.83	22.91	76.5	14.54	PM7
BDSe-4Br	[37]	1.39	−5.65/−3.96	0.83	22.12	71.5	13.12	PM7
C8T-BDT4Cl	[38]	1.40	−5.75/−3.92	0.86	19.65	72.3	12.21	PM6
C8T-BDSe4Cl	[38]	1.39	−5.71/−3.96	0.85	21.79	72.7	13.50	PM6
SRID-4F	[39]	1.44	−5.52/−3.90	0.85	20.21	75.2	13.05	PBDB-T-2F
TRID-4F	[39]	1.48	−5.52/−3.90	0.89	18.45	75.0	12.33	PBDB-T-2F
TTTIC	[40]	1.58	−5.69/−3.90	0.94	18.29	70.1	12.05	PM6
TSeTIC	[40]	1.53	−5.65/−3.91	0.93	19.42	75.9	13.71	PM6
TSeIC-M-Cl	[41]	1.47	−5.64/−4.01	0.82	16.10	73.6	9.72	PBT1-C
TSeIC-P-Cl	[41]	1.47	−5.62/−3.98	0.85	18.67	71.0	11.26	PBT1-C
SeCT-IC	[42]	1.54	−5.53/−3.92	0.89	18.09	67.7	10.89	J71
CSe-T-IC	[42]	1.58	−5.60/−3.91	0.91	17.17	54.8	8.52	J71
CTSe-IC	[42]	1.56	−5.55/−3.90	0.92	18.21	68.8	11.59	J71
BPF-4F	[43]	1.36	−5.58/−3.98	0.85	22.10	67.4	12.60	SZ5
BPT-4F	[43]	1.36	−5.59/−4.00	0.85	24.80	79.1	16.80	SZ5
BPS-4F	[43]	1.29	−5.54/−4.00	0.82	25.40	77.9	16.30	SZ5
CH1007	[44]	1.30	−5.59/−3.97	0.82	27.48	75.6	17.08	PM6
S-YSS-Cl	[45]	1.34	−5.71/−3.85	0.86	25.85	75.3	16.73	PM6
A-WSSe-Cl	[45]	1.31	−5.70/−3.86	0.85	26.58	77.5	17.51	PM6
S-WSeSe-Cl	[45]	1.30	−5.66/−3.88	0.83	26.35	73.4	16.01	PM6
YSe-C3	[46]	1.36	−5.65/−4.29	0.83	23.36	60.0	11.63	PM6
YSe-C6	[46]	1.32	−5.67/−4.35	0.85	25.94	73.0	16.11	PM6
YSe-C9	[46]	1.33	−5.65/−4.32	0.85	23.21	72.0	14.20	PM6
mBzS-4F	[47]	1.25	−5.61/−3.92	0.80	27.72	76.4	17.02	PM6
PN6SBO-4F	[47]	1.28	−5.63/−3.87	0.83	23.13	66.7	12.73	PM6
AN6SBO-4F	[47]	1.27	−5.60/−3.88	0.82	16.06	63.0	8.32	PM6
EHN6SEH-4F	[47]	1.29	−5.59/−3.89	0.81	28.83	74.6	17.48	PM6
EHBzS-4F	[48]	1.24	−5.61/−3.86	0.83	27.58	70.1	15.94	PM6
Y6Se	[49]	1.32	−5.70/−4.15	0.84	27.55	75.3	17.70	D18
Y6-2Se	[50]	1.34	−5.58/−3.84	0.84	24.81	71.0	14.94	PM6
BTP-Se	[51]	1.24	−5.64/−4.04	0.71	28.66	69.7	14.20	PBDB-T
4TIC	[52]	1.40	−5.28/−3.87	0.82	19.13	65.0	10.20	PBDB-T
STIC	[52]	1.32	−5.20/−3.91	0.77	19.96	63.0	9.68	PBDB-T
MQ3	[53]	1.37	−5.63/−3.94	0.91	22.19	66.9	13.51	PM6
MQ5	[53]	1.34	−5.59/−3.95	0.86	24.48	74.3	15.64	PM6
MQ6	[53]	1.35	−5.61/−3.95	0.88	24.62	75.7	16.39	PM6
ITCPTC-Se	[54]	1.59	−5.65/−4.00	0.87	15.20	68.3	9.02	PBDB-T
IDT2Se	[55]	1.45	−5.41/−3.87	0.89	17.49	60.8	9.36	PBDB-T
IDT2Se-4F	[55]	1.39	−5.51/−4.00	0.79	21.49	65.9	11.19	PBDB-T
IDT2SeC2C4	[56]	1.44	−5.30/−3.91	0.88	17.18	58.9	8.92	PBDB-T
IDT2SeC2C4-2F	[56]	1.40	−5.37/−3.99	0.81	19.19	65.6	10.20	PBDB-T
IDT2SeC2C4-4F	[56]	1.30	−5.50/−4.08	0.77	22.02	62.3	10.57	PBDB-T
IDTO-T-4F	[30]	1.45	−5.49/−3.88	0.86	20.12	72.7	12.62	PBDB-T
IDTO-Se-4F	[30]	1.40	−5.48/−3.90	0.83	18.55	69.2	10.67	PBDB-T
IDTO-TT-4F	[30]	1.38	−5.39/−3.89	0.86	17.21	69.4	10.21	PBDB-T
MPU1	[57]	1.82	−5.81/−3.99	0.95	12.67	60.0	7.22	SM1
MPU4	[57]	1.69	−5.65/−3.96	0.99	14.91	64.0	8.96	SM1
2T2Se-F	[58]	1.31	−5.52/−3.83	0.88	20.63	66.9	12.17	PM6
PFY-0Se	[59]	1.51	−5.68/−3.88	0.90	20.90	68.8	13.00	PBDB-T
PFY-1Se	[59]	1.50	−5.68/−3.89	0.90	21.20	72.9	13.80	PBDB-T
PFY-2Se	[59]	1.45	−5.64/−3.91	0.88	23.40	72.0	14.70	PBDB-T
PFY-3Se	[59]	1.45	−5.65/−3.92	0.87	23.60	73.7	15.10	PBDB-T
PY2S-H	[60]	1.46	−5.76/−3.87	0.94	22.30	70.7	14.80	PM6
PY2S-F	[60]	1.43	−5.78/−3.91	0.92	23.30	70.5	15.10	PM6
PY2Se-F	[60]	1.42	−5.76/−3.93	0.89	24.40	72.2	15.60	PM6
PY2Se-Cl	[60]	1.42	−5.75/−3.93	0.89	24.50	74.3	16.10	PM6
P(NDI2OD-Se-Th0)	[61]	1.61	−5.78/−4.17	0.86	13.23	59.0	6.88	PBDB-T
P(NDI2OD-Se-Th0.5)	[61]	1.55	−5.65/−4.11	0.87	13.35	60.0	7.01	PBDB-T
P(NDI2OD-Se-Th0.7)	[61]	1.52	−5.61/−4.09	0.87	13.73	63.0	7.69	PBDB-T
P(NDI2OD-Se-Th0.8)	[61]	1.49	−5.55/−4.06	0.88	14.20	66.0	8.30	PBDB-T
P(NDI2OD-Se-Th0.9)	[61]	1.47	−5.52/−4.04	0.88	14.44	63.0	8.20	PBDB-T
P(NDI2OD-Se-Th1.0)	[61]	1.45	−5.48/−4.03	0.89	13.63	60.0	7.39	PBDB-T

**Table 2 materials-15-07883-t002:** OSC device parameters of selenophene donors.

Donor	Ref.	E_g_^opt^(eV)	HOMO/ LUMO (eV)	Voc	J_max_/ mA^−2^	FF (%)	PCE (%)	Acceptor
L1	[66]	1.76	−5.32/−3.51	0.83	25.28	69.8	14.6	Y6
L2	[66]	1.77	−5.33/−3.53	0.83	26.35	72.1	15.80	Y6
VC6	[67]	1.72	−5.44/−3.72	0.82	14.31	62.0	7.27	PC_71_BM
VC7	[67]	1.40	−5.30/−3.90	0.89	15.98	66.0	9.24	PC_71_BM
PCDSeBT	[68]	1.70	−5.40/−3.70	0.79	11.70	45.0	4.12	PC_71_BM
PSeBDDIDT	[69]	1.76	−5.40/−3.64	0.90	16.04	60.0	8.65	PC_71_BM
PTzBDDIDT	[69]	1.86	−5.67/−3.81	0.97	8.12	35.0	2.75	PC_71_BM
PBPD-Th	[70]	1.90	−5.42/−3.36	0.95	12.40	71.0	8.40	PC_71_BM
PBPDSe	[70]	1.77	−5.35/−3.31	0.90	14.90	73.0	9.80	PC_71_BM
PhI-Se	[71]	2.08	−5.61/−3.65	0.85	26.30	76.8	17.20	PC_71_BM
PSeTD1	[72]	1.75	−5.34/−2.70	0.83	10.92	68.7	6.22	PC_71_BM
PSeTD2	[72]	1.75	−5.31/−2.74	0.85	13.55	71.1	8.18	PC_71_BM
P(Se)	[73]	1.67	−5.49/−3.82	0.86	10.24	60.0	5.30	PC_71_BM
PTEI-S	[74]	1.52	−5.29/−3.70	0.87	8.94	55.0	4.28	PC_61_BM
PSEI-T	[74]	1.53	−5.23/−3.66	0.84	6.59	61.0	3.36	PC_61_BM
PBDTT-SeDPP	[75]	1.38	−5.25/−3.70	0.69	16.80	62.0	7.20	PC_71_BM
PDTP-BDTSe	[76]	1.80	−5.46/−3.62	0.85	14.80	56.3	7.08	PC_71_BM
PBDTSe-BT	[77]	1.68	−4.98/−2.98	0.66	6.58	55.3	2.39	PC_71_BM
PBDTSe-FBT	[77]	1.66	−5.01/−3.04	0.67	7.23	34.6	1.68	PC_71_BM
PBDTSe-FFBT	[77]	1.68	−5.05/−3.07	0.72	7.24	50.6	2.63	PC_71_BM
PBDTSe-TT	[78]	1.40	−5.19/−3.25	1.64	8.80	69.0	9.90	PC_71_BM
PBDT-Se-TAZ	[79]	1.92	−5.23/−3.43	0.81	18.63	66.7	10.07	ITIC
PBDTS-Se-TAZ	[79]	1.90	−5.29/−3.52	0.84	19.51	75.1	12.31	ITIC
PBDT(F)TPD	[80]	1.78	−4.85/−2.76	0.90	7.80	42.0	3.00	PC_71_BM
PBDT(T)TPD	[80]	1.88	−5.01/−2.76	1.0	11.10	58.0	6.50	PC_71_BM
PBDT(S)TPD	[80]	1.85	−4.98/−2.76	0.98	8.70	48.0	4.70	PC_71_BM
PIDSe-DFBT	[81]	1.80	−4.64/−2.84	0.89	13.70	56.3	6.79	PC_71_BM
J75	[82]	1.93	−5.49/−3.66	0.96	17.11	69.5	11.41	m-ITIC
J76	[82]	1.86	−5.41/−3.66	0.91	17.04	71.2	11.04	m-ITIC

## Data Availability

Not applicable.

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
