# Peer review of "Recent Advances in Selenophene-Based Materials for Organic Solar Cells"

_materials, 2022, doi:10.3390/ma15227883_

Round 1
Reviewer 1 Report
Authors summarized the use of selenophene in organic solar active layer materials successfully which I believe an important contribution to the organic solar cell literature. They organized their paper as selenophene acceptors for organic solar cells followed by selenophene donors in organic solar cells which is a good way to put forth. I recommend the publication of paper after minor revision.
Abstract should be improved to include a summary of paper such as utilizing selenophene as donor and acceptor.
DFT method for “Figure 2. DFT for furan, thiophene and selenophene” should be given.
Story telling should be improved especially in the pages 8-14. It is hard to read the review with all these structures and author names are given in succession. Increasing the number of paragraphs and decreasing the number of author and structure names can be helpful for the audience to follow paper.
Better use of English language is required such as:
“planarity” should be used instead of “flatness” in conclusion.
"At the same time," can be replaced by "In addition"
the words "good", "easy" can be replaced by scientific synonyms.
74 references for such a comprehensive review is weak
Recent studies for “selenophene-based materials for organic solar cells” should be included such as:
https://doi.org/10.1021/acsenergylett.0c02230
https://doi.org/10.1016/j.electacta.2021.139298
https://doi.org/10.1039/C9TA02415H
Author Response
Reviewer1 Authors summarized the use of selenophene in organic solar active layer materials successfully which I believe an important contribution to the organic solar cell literature. They organized their paper as selenophene acceptors for organic solar cells followed by selenophene donors in organic solar cells which is a good way to put forth. I recommend the publication of paper after minor revision. Abstract should be improved to include a summary of paper such as utilizing selenophene as donor and acceptor.
Response: Thank you very much for your reminder. We added “This paper introduces the organic solar active layer materials containing selenium benzene in recent years, which can be simply divided into donor materials and acceptor materials. Replacing sulfur atoms with selenium atoms in these materials can effectively reduce the corresponding optical band gap of materials, improve the mutual solubility of donor recipient materials, and ultimately improve the device efficiency” at the end on page
- DFT method for “Figure 2. DFT for furan, thiophene and selenophene” should be given. Response: Thank you very much for your reminder. We added “The geometry of all molecules and complexes was optimized through density functional theory (DFT). All the DFT computations were performed by the B3LYP density functional method. The 6–31G(d) basis set was used for the energy calculation of the molecules. All these calculations were performed with Gaussian 16 software package.” at the end on page
- 2. Story telling should be improved especially in the pages 8-14. It is hard to read the review with all these structures and author names are given in succession. Increasing the number of paragraphs and decreasing the number of author and structure names can be helpful for the audience to follow paper. Response: Thank you very much for your reminder. On page 8-14, we modified the multiple occurrences of names and divided the paragraphs. Better use of English language is required such as: “planarity” should be used instead of “flatness” in conclusion. "At the same time," can be replaced by "In addition" the words "good", "easy" can be replaced by scientific synonyms.
- Response: Thank you very much for your reminder. We have revised some words, such as the planarity on pages 2,6,8,13,17,19, good on pages 1,2,3, at the same time on pages 9,11,18, etc. 74 references for such a comprehensive review is weak Recent studies for “selenophene-based materials for organic solar cells” should be included such as: https://doi.org/10.1021/acsenergylett.0c02230 https://doi.org/10.1016/j.electacta.2021.139298 https://doi.org/10.1039/C9TA02415H Response: Thank you very much for your reminder. We have written an overview of the first linked article on page 11, line 10. The second and third connected articles are summarized on pages 16 and 18 of the article. We added more ref, and 82 in total in the last revision.
Reviewer 2 Report
In this manuscript, Liu et al. reviewed some performances related to the use of selenophene-based materials for efficient organic solar cells. Therefore, before proceeding further, the authors need to justify many things to make this review at the required level of publication. Here below are some of the comments that the authors should address to improve this review article's quality.
1. Already there is a vast number of reviews on this topic. The authors should point out why there is a need for another review. What are the new/novel points in their review?
2. A detailed in-depth discussion of the photophysics of these types of solar cells (for example, the interplay between charge transfer and energy transfer, the kinetics of different photogenerated species like exciton, CT-state, excimer, etc.) should be discussed.
3. The format of the figures is not constant, and the quality is not good, one cannot read the figure captions.
4. Figure 3, 4, and 5 have very poor visualization. This needs to be improved.
5. Author claimed that with these materials’ device efficiency could be increased up to 20% but, there is no motivation and strategies present that how to get 20% PCE with these materials.
6. Most of the references are very old, therefore it didn’t have recent advancements in the field.
7. Finally, the quality of English should be improved.

Author Response
Revierwer2 In this manuscript, Liu et al. reviewed some performances related to the use of selenophene-based materials for efficient organic solar cells. Therefore, before proceeding further, the authors need to justify many things to make this review at the required level of publication. Here below are some of the comments that the authors should address to improve this review article's quality. 1. Already there is a vast number of reviews on this topic. The authors should point out why there is a need for another review. What are the new/novel points in their review?
Response: Thank you very much for your reminder. We think that in our article, we mainly divided the structure of selenium containing non fullerene electron receptors in detail, and on this basis, we mainly analyzed and sorted out the UV absorption and surface characterization of some materials. Some selenium containing materials show obvious ultraviolet absorption red shift and interpenetrating nanofiber structure, which are described in the article.
2. A detailed in-depth discussion of the photophysics of these types of solar cells (for example, the interplay between charge transfer and energy transfer, the kinetics of different photogenerated species like exciton, CT-state, excimer, etc.) should be discussed. Response: Thank you very much for your reminder. We have supplemented it on page 9.
3. The format of the figures is not constant, and the quality is not good, one cannot read the figure captions. Response: Thank you very much for your reminder. We have changed the format of the picture on pages 2, 3, 4, 5, 6, 13 and 15 respectively.
4. Figure 3, 4, and 5 have very poor visualization. This needs to be improved. Response: Thank you very much for your reminder. We changed the image and adjusted the size and order of each molecule to make the structure of each molecule clearer.
5. Author claimed that with these materials’ device efficiency could be increased up to 20% but, there is no motivation and strategies present that how to get 20% PCE with these materials. Response: Thank you very much for your reminder. We expanded the description of material A-WSSe Cl on page 11.
6. Most of the references are very old, therefore it didn’t have recent advancements in the field. Response: Thank you very much for your reminder. We have added 8 new materials to the article.
7. Finally, the quality of English should be improved. Response: Thank you very much for your reminder. We have replaced some words and corrected some grammar, such as pages 2 and 8, and we carefully revised though the whole paper.
Reviewer 3 Report
The manuscript ID: materials-2009973 “Recent advances in selenophene-based materials for organic solar cells”. In this review article, the author focused on various selenophene based organic solar cells. The novelty is unclear in the current form of the review article. It seems to be unable to provide an outlook of the further development of new solar cells with better performances. Their presentation on review writing was not so exciting. I am recommending this review because of only for few articles reported on selenophene-based materials. So, the topic is interesting for the referee. Therefore, I recommend publication only after minor revisions.
1) The abstract of the review article is not so exciting, the author needs to rewrite the abstract.
2) Page No 1; In Introduction; Since American scientists, Kearns and Calvin first prepared organic photovoltaic devices---- the author needs to include references.
3) Page No 1; In the Introduction; the author has written “At present, researchers have focused on the research of the third generation solar cells.” my question is, “Most of the 3rd generation technologies are not yet commercially implemented” why? The author needs to include the proper reason in the review article.
4) Page No 2: the author needs to adjust figure 1 in between two paras.
5) I didn’t find the advantages of selenophene for small molecules. The author needs to include it in the manuscript. Are they any drawbacks with selenophene-based materials, if the author needs to include them in this review article
6) Page No 3; In figure 2: I think that the author needs to include an optimized structure for furan, thiophene and selenophine.
7) In furan, thiophene and selenophine, which one is the best for OSCs and why? Why not Se?
8) Which one is effective for ICT (needs to include in page No:6 in 2.1.1--)
9) Page No:13; white the correct form for C60.
10) Mainly missing references, please include proper reference for better understanding. (for example: page No2; the author has written. “Active layer materials, ---- improve the efficiency of the device”.
11) Please verify each sentence of this paper, Some typo errors. Especially space in between two words.
Author Response
Reviewer3
The manuscript ID: materials-2009973 “Recent advances in selenophene-based materials for organic solar cells”. In this review article, the author focused on various selenophene based organic solar cells. The novelty is unclear in the current form of the review article. It seems to be unable to provide an outlook of the further development of new solar cells with better performances. Their presentation on review writing was not so exciting. I am recommending this review because of only for few articles reported on selenophene-based materials.
So, the topic is interesting for the referee. Therefore, I recommend publication only after minor revisions. 1) The abstract of the review article is not so exciting, the author needs to rewrite the abstract.
Response: Thank you very much for your reminder. We have expanded and modified the abstract.
2) Page No 1; In Introduction; Since American scientists, Kearns and Calvin first prepared organic photovoltaic devices---- the author needs to include references. Response: Thank you very much for your reminder. We have added the corresponding references in the article, on page 1. [2] Kearns D, Calvin M, Photovoltaic effect and photoconductivity in laminated organic systems, The Journal of Chenical Physics, 1958,29(4), 950-951
3) Page No 1; In the Introduction; the author has written “At present, researchers have focused on the research of the third generation solar cells.” my question is, “Most of the 3rd generation technologies are not yet commercially implemented” why? The author needs to include the proper reason in the review article. Response: Thank you very much for your reminder. We apologize for the unclear expression of the article, and we added it in the introduction section on the first page.
4) Page No 2: the author needs to adjust figure 1 in between two paras. Response: Thank you very much for your reminder. We are sorry for the format error of Figure 1. We have performed l on its location and format.
5) I didn’t find the advantages of selenophene for small molecules. The author needs to include it in the manuscript. Are they any drawbacks with selenophene-based materials, if the author needs to include them in this review article Response: Thank you very much for your reminder. We apologize for the lack of description of the benefits of selenophene on small molecules, which we added on page 3. The disadvantages of selenophene containing materials are described on the last page 21. “However, the materials containing selenophene… …”
6) Page No 3; In figure 2: I think that the author needs to include an optimized structure for furan, thiophene and selenophine. Response: Thank you very much for your reminder. We are very sorry for the lack of pictures. We have improved and optimized the pictures, which are shown on page 3.
7) In furan, thiophene and selenophine, which one is the best for OSCs and why? Why not Se? Response: Thank you very much for your reminder. We are very sorry for the lack of the elaboration part of the article. We have supplemented it on page 2. “Although the existence of selenium”
8) Which one is effective for ICT (needs to include in page No:6 in 2.1.1--) Response: Thank you very much for your reminder. We are sorry for the lack of ICT description. We have expanded it on page 6.
9) Page No:13; white the correct form for C60. Response: Thank you very much for your reminder. We are very sorry for the format error. We have corrected the format.
10) Mainly missing references, please include proper reference for better understanding. (for example: page No2; the author has written. “Active layer materials, ---- improve the efficiency of the device”. Response: Thank you very much for your reminder. We apologize for the lack of references, and we have supplemented them. [23] Aga F G, Bakare F F, Dibaba S T, et al., Investigation of the Impact of Active Layer and Charge Transfer Layer Materials on the Performance of Polymer Solar Cells through Simulation, Advances in Materials Science and Engineering, 2022, 2022(1), 1-7 [24] Amollo T A, Mola G T and Nyamori V O, Organic solar cells: Materials and prospects of graphene for active and interfacial layers, Critical Reviews in Solid State and Materials Sciences, 2019, 45(4), 261-288
11) Please verify each sentence of this paper, Some typo errors. Especially space in between two words. Response: Thank you very much for your reminder. We are very sorry for the mistakes in English grammar. We have adjusted the spaces, words and grammar on pages 8, 9, 13, 20 and 21.
Round 2
Reviewer 2 Report
Thanks for the revision.
Author Response
Dear Reviewer and Editor,
We thank again for your kind comments and we made revision as following:
- Some grammar mistakes and typos were corrected as marked in red.
- We modified the DFT figure to show which symbol represents which chemical mentioned in the caption in the new revision.
- Padge 10, "Someone replaced the sulfur atom near the phenyl, and someone replaced the sulfur on the outermost thiophene." was replaced by the following to be clear:
Hence, the sulfur atom on thiophene near the central phenyl unit was replaced by sele-nium to construct molecules such as C8T-BDSe-4Cl. On the other hand, the sulfur atom of thiophene far from the central phenyl was substituted by selenium to design molecules such as SRID-4F.
